# Co-Assembly of 40S and 60S Ribosomal Proteins in Early Steps of Eukaryotic Ribosome Assembly

**DOI:** 10.3390/ijms20112806

**Published:** 2019-06-08

**Authors:** Jesse M. Fox, Rebekah L. Rashford, Lasse Lindahl

**Affiliations:** Department of Biological Sciences, UMBC, 1000 Hilltop Circle, Baltimore, MD 21250, USA; foxjessem@gmail.com (J.M.F.); rashford@princeton.edu (R.L.R.)

**Keywords:** ribosome assembly, ribosomal subunit co-assembly, rRNA, ribosomal protein, protein synthesis

## Abstract

In eukaryotes three of the four ribosomal RNA (rRNA) molecules are transcribed as a long precursor that is processed into mature rRNAs concurrently with the assembly of ribosomal subunits. However, the relative timing of association of ribosomal proteins with the ribosomal precursor particles and the cleavage of the precursor rRNA into the subunit-specific moieties is not known. To address this question, we searched for ribosomal precursors containing components from both subunits. Particles containing specific ribosomal proteins were targeted by inducing synthesis of epitope-tagged ribosomal proteins followed by pull-down with antibodies targeting the tagged protein. By identifying other ribosomal proteins and internal rRNA transcribed spacers (ITS1 and ITS2) in the immuno-purified ribosomal particles, we showed that eS7/S7 and uL4/L4 bind to nascent ribosomes prior to the separation of 40S and 60S specific segments, while uS4/S9, uL22, and eL13/L13 are bound after, or simultaneously with, the separation. Thus, the incorporation of ribosomal proteins from the two subunits begins as a co-assembly with a single rRNA molecule, but is finished as an assembly onto separate precursors for the two subunits.

## 1. Introduction

The biogenesis of eukaryotic ribosomes is a massive undertaking involving binding of ribosomal proteins (r-proteins) to rRNA and processing of precursor rRNA along pathways that are facilitated by over 250 ribosomal assembly factors [1,2]. The process begins in the nucleolus with RNA polymerase I transcribing a long precursor rRNA (“35S” in *Saccharomyces cerevisiae* (yeast) and “45–47S” in mammalian cells) that includes sequences for three of the four mature rRNAs (18S rRNA, 5.8S, and 25S) in addition to “Transcribed Spacers” (Figure 1A) [3]. The External Transcribed Spacers (5′ and 3′ ETS) make up the 5′ and 3′ parts of the primary transcript, while the Internal Transcribed Spacers (ITS1 and ITS2) are interstitial between the 18S, 5.8S, and 25S rRNA parts. As it is transcribed, the precursor transcript associates with a large number of ribosomal assembly factors to form the “90S” precursor particle, which is subsequently split into subunit-specific entities by cutting the precursor rRNA at the A2 site in ITS1 (Figure 1A). In rapidly growing yeast cells 70%–80% of nascent precursor rRNA molecules are cleaved while transcription of the rRNAs for the large ribosomal subunit is still ongoing [4,5,6]. Subsequent to the ITS1 cleavage, the 40S and 60S are completed along separate multiple-step pathways [1,2]. Ribosomal proteins (r-proteins) are made in the cytoplasm concurrently with the transcription and processing of the rRNA and then chaperoned to the nucleolus or nucleus where they associate with the nascent ribosomal subunits in a hierarchical fashion [7,8,9].

Over the last couple of decades, the ribosome assembly process has been elucidated in very significant detail [1,2,10,11,12]. Even though rRNAs for both subunits are co-transcribed, the complexity of the ribosome construction has necessitated separate exploration of the assembly of the two subunits. Incomplete ribosomal precursor particles for each of the subunits have been purified and analyzed for structure and content of rRNA and assembly factors. However, this approach does not clarify the relative timing of association of r-proteins with the ribosomal precursor particles, and cleavage of the primary rRNA transcript in the physiological situation where the assembly of the two subunits occurs simultaneously. Of special interest is the question of whether 60S and 40S proteins are co-assembled before the long rRNA precursor is cleaved and the ribosomal assembly is separated into subunit-specific pathways. Furthermore, determining the content of r-proteins in the precursor subunits is complicated by the fact that the number of mature ribosomes vastly outnumbers assembly intermediates, which may result in transfer of r-proteins from mature ribosomes to assembly intermediates during cell lysis and purification of nascent ribosomes [12]. 

Challenged by these questions, we developed an approach to determine the relative timing of addition of r-proteins to, and the cleavage of, the pre-rRNA within ITS1 (Figure 1A). We focused on the 40S proteins uS4/S9 and eS7/S7, as well as the 60S proteins uL4/L4 and eL13/L13 which are all known to be incorporated early in the assembly before their respective ribosomal subunits migrate from the nucleolus to the nucleus [7]. Hemagglutinin (HA)-tagged versions of the r-proteins were expressed for a short time and precursor particles isolated using anti-HA. Identification of r-proteins and transcribed spacers in the immuno-precipitate showed that eS7 and uL4 associate with precursor ribosomes prior to the separation of 40S and 60S specific segments, while uS4/S9, uL22, and eL13 bind after, or simultaneously with, ITS1 cleavage

## 2. Results

### 2.1. Induction of HA-Tagged r-Protein Synthesis

To differentiate recently synthesized r-proteins from bulk r-proteins, we used β-estradiol to induce the synthesis of hemagglutinin (HA)-tagged r-proteins (Figure 1B (i)). Genes for N-terminally HA-tagged r-proteins expressed from a *GAL::CYC* hybrid promoter [13] were constructed on plasmids and introduced into a strain that harbors a chromosomal gene for a composite transcription factor built from the *GAL4* DNA binding domain, a glucocorticoid responsive element, and the viral VP16 RNA polymerase activation domain [14] (Figure 1B (ii)). Since the proteins are first incorporated into precursor particles and only later become parts of mature ribosomes, short induction times should assure that a large fraction of the tagged r-proteins is harbored in precursor ribosomes (pre-ribosomes), thus ameliorating any problems with artificial transfer of HA-r-proteins from mature ribosomes to pre-ribosomes during lysis and purification. Western blots probed with anti-HA showed that the level of the HA-tagged r-proteins began increasing 5–10 min after addition of β-estradiol (Figure 1C). Prior to induction there was a low basal level of HA-protein, as expected, since the β-estradiol-inducible promoter is somewhat leaky [15]. 

### 2.2. HA-Tagged r-Proteins Are Incorporated into Ribosomal Precursor Particle before Migrating into Mature Ribosome 

We followed the incorporation of tagged r-proteins into ribosomal particles using sucrose gradient fractionation of lysates harvested 0, 10, 15, 30, and 60 min after induction. Western analysis of the sucrose gradient fractions showed that after 10 min of induction, HA-uL4 was predominantly seen in the 60S fractions, presumably corresponding to the 66S-type particles previously described (Figure 2A) [3]. The paucity of tagged protein in 80S bore out our prediction that most HA-r-proteins would be in precursor particles after short induction times. Traces of HA-uL4 that sediment at approximately 40S suggest the existence of short-lived early precursor particles for the 60S that have not been described. Between 30 and 60 min, tagged uL4 was building up in the 80S peak and polysomes, showing that HA-uL4 containing ribosomal precursors were matured into functional 60S subunits. Unexpectedly, we also saw HA-uL4 in material close to the top of the gradient after 30–60 min of induction (Figure 2A). We did not examine these complexes, but the timing of their appearance suggests that they are not ribosomal precursor particles. Rather, we suggest that they contain protein synthesized in excess of other ribosomal components. We also followed the incorporation of HA-eS7 into 40S subunits, 80S, and polysomes (Figure 2B). After 15 min of induction, the protein was seen in the 40S and 80S peak, and by 30 and 60 min it was also seen in polysomes. As was the case with tagged uL4, the 40S protein eS7 also accumulated outside of ribosomes in complexes sedimenting towards the top of the gradient. Together these results suggest that the tagged proteins are incorporated sequentially into ribosomal precursor particles and functional ribosomes like wildtype r-proteins. However, protein synthesized in excess of other ribosomal components appears to form aberrant complexes after longer induction periods.

### 2.3. Co-Precipitation of Other r-Proteins with HA-Tagged r-Proteins

If r-proteins associate with precursor ribosomal particles before separation of the 40S and 60S precursors, it might be expected that some early assembly intermediate(s) contain proteins from both subunits. To look for such complexes, we immunoprecipitated ribosomal particles using anti-HA coated beads and analyzed for co-precipitating r-proteins by Western blotting. We first performed an experiment with a strain in which HA-uL4 was expressed from the β-estradiol-inducible promoter. Precipitating from lysates prepared 30 min after induction brought down HA-uL4, wildtype L4 and uL22 (Figure 3A). This was expected, because the sucrose gradients indicate that by 30 min HA-uL4 had migrated into mature 60S particles (Figure 2A). Moreover, small amounts of the same proteins were observed in precipitates from the same strain prepared before addition of β-estradiol (t = 0), commensurate with the low level of basal expression of HA-uL4.

Induction of the strain carrying HA-eS7 under β-estradiol-inducible promoter generated novel information about co-assembly of 40S and 60S proteins. Ribosomal particles were pulled down with anti-HA beads before and 30 min after induction of HA-eS7 synthesis. The Western blot showed that uL4, but not uL22, co-purified with HA-eS7 (Figure 3B). Since eS7 was found in 80S and polysomes after 30 min of induction (Figure 2B), we had expected that both 80S and polysomes should have been precipitated with anti-HA, i.e. both uL4 and uL22 were expected to copurify with HA-eS7. In other words, there is a contrast between the absence of uL22 in the eS7 precipitate and the very efficient pull-down of uL22 with uL4. We hypothesize that the absence of uL22 in the HA-eS7 pull-down is caused by a failure to precipitate 80S containing HA-eS7. This argument is supported by comparing the structures of the 90S precursor ribosomes [10] and mature 80S ribosomes [16] (Figure 4). In the 90S, the eS7 N-terminus is positioned completely clear of other ribosomal components that could prevent access of the anti-HA. However, in the 80S, the 60S protein eL19 comes close to the eS7 N-terminus. Since HA-eS7 is elongated by nine amino acids that make up the HA-tag, it is plausible that eL19 could obstruct the access of the HA antibody on agarose beads. We note that the HA-epitope on eS7 is intact, since it reacts with anti-HA in Western blots of proteins that have been extracted from the 80S ribosomes (Figure 1B). We therefore interpret the results of the pull-down experiments to mean that uL4 in the eS7 exist together in a precursor particle that is upstream of the incorporation of uL22. Hence, uL4 and eS7 must be added to pre-ribosomes before the 40S and 60S specific rRNA moieties are separated by cleavage in ITS1, while uL22 is added after. 

### 2.4. Co-Purification of Internal rRNA Spacers with HA-Tagged uS4 and uL4

To test the hypothesis that uL4 binds to the ribosomal assembly complex before A2 cleavage, we examined if ITS1 and ITS2 were co-precipitated with tagged ribosomal proteins. These sequences are hallmarks that distinguish ribosomal precursor ribosomes from mature ribosomes, because they are removed and degraded during ribosome maturation. Furthermore, ITS1 and ITS2 coexist in early ribosomal precursor particles, but are separated into separate particles by the cleavage at A2 in ITS1 (Figure 1A). Thus, ITS1 and ITS2 can be used to coordinate rRNA processing with incorporation of the HA-tagged r-proteins. RNA was extracted from the anti-HA precipitate and loaded onto a nylon membrane in a slot pattern. As controls we used total RNA purified from lysates prior to the immunoprecipitation as well as RNA from immune precipitates of lysates made from the host strain not containing a plasmid-borne HA-tagged r-protein gene. The blot was then probed with ^32^P end-labeled oligonucleotides with sequences complementary to the *S. cerevisiae* ITS1 upstream of A2 (O1663) and ITS2 upstream of C2 (O1660) (Figure 1A), which serve to identify spacer sequences linked to the 18S rRNA in pre-40S and the 5.8S rRNA in pre-60S, respectively, after cleavage at A2. If a given HA-r-protein is added prior to A2 cleavage, both ITS1 upstream of A2 and ITS2 will be associated with the particles pulled down with the HA-tagged protein. If the HA-tagged r-protein is added after cleavage, only one of the spacer elements should co-precipitate with the HA-r-protein. Importantly, since the fraction of precursor particles containing the HA-tagged r-proteins increases after induction, the efficiency of co-precipitation of the transcribed spacer should also increase after induction of the HA-tagged proteins. Thus, an increase of ITS1 or ITS2 in the precipitate indicated that the respective transcribed spacer is co-precipitated, because it is part of the ribosomal precursor particle(s). If the content of a particular transcribed spacer does not increase after induction, it must be precipitated non-specifically.

We determined the content of internal transcribed spacer elements in ribosomal particles pulled down after induction of HA-uS4, HA-uL4, or HA-eL13, all proteins that are added to the nascent ribosomal subunits in the nucle(o)lus prior to nuclear export [7]. We note that the N-terminal end of uL4 is available without obvious obstruction in models of early and late (cytoplasmic) precursor 60S as well as in mature 80S ribosomal particles (Figure 4). Unfortunately, the 90S precursor structure model does not include the N-terminal end of uS4, so we cannot compare this aspect of uS4 in the 90S and 80S. Autoradiographic bands were seen for both the ITS1 and the ITS2 probes with the pull down of HA-uL4 and HA-uS4, but for eL13, we only saw bands with ITS 2 (Figure 5A). This suggests that eL13 is added after A2 cleavage. To determine if the intensity of the ITS1 and ITS2 bands increased with induction of HA-uL4 or HA-uS4, as expected if the ITS parts co-precipitate with the tagged proteins in precursor particles, we quantified the bands using ImageJ. However, the quantifications of bands generated with the two probes were not directly comparable, since the specific activity of the two oligonucleotides varied due to differences between the labeling efficiencies. We therefore normalized the raw data obtained for the pull-down material to the intensity in slots loaded with total cellular RNA and subtracted the background signal from precipitates coming for the control strain without an HA tagged r-protein gene. As shown in Figure 5B (left), the signal intensity for ITS2 increased after 10–15 min of HA-uL4 induction, indicating that HA-uL4 binds to its rRNA target prior to eliminating ITS2. Very significantly, the intensity of the bands formed with O1663, targeting ITS1 upstream of A2, also increased 10–15 min after induction of HA-uL4, i.e. at a time when little, if any, HA-uL4 is seen in the 80S peak (Figure 5B, left), showing that HA-uL4 also co-precipitated ITS1 upstream of A2. Thus, the HA-uL4 precipitate contained both ITS1 upstream of A2 and ITS2, indicating that uL4 associates with its target in rRNA prior to A2 cleavage, as was also concluded from the finding of uL4 in the eS7 immuno-precipitate (Figure 3B). By 30 min the ITS1 and ITS2 content in the precipitated material decreased as would be expected, because an increasing fraction of the tagged proteins is present in mature ribosomes as the induction time increases. Although bands were also seen for both ITS1 and ITS2 in the pull-down with uS4 (Figure 5B, right), the pattern of the quantification of the bands differed from the experiment with pull down of uL4. Only the content of ITS1 increased, while the ITS2 content did not. This suggests that co-precipitation of ITS1 with HA-uS4 is due to ITS1 being part of a uS4-containing precursor particle, but that the ITS2 in the HA-uS4 precipitate must be due to unspecific binding. Overall, the co-precipitation of transcribed spacer sequences thus suggests uS4 binds to the pre-40S after A2 cleavage.

The interpretation of our co-precipitation experiments is based on the assumption that HA-tagged r-proteins are not exchanged between ribosomal particles after lysis, which could result in artificial association of the tagged proteins with precursor particles. To test this assumption, we used *Kluyveromyces lactis* as an internal control for post-lysis transfer. The uL4 sequence of *K. lactis* is identical to uL4 of *S. cerevisiae*, and the 25S rRNA sequences in the two species are 99% identical. If HA-uL4 is exchanged between ribosomal particles in the lysate, some of the HA-tagged protein should bind to precursor ribosomal particles from *K. lactis* and therefore be associated with internal transcribed sequences from this species. To specifically identify ITS1 and ITS2 transcripts from *K. lactis*, we exploited differences between the internal transcribed spacers in *S. cerevisiae* and *K. lactis* to design a set of oligonucleotide probes that hybridize to the internal transcribed spacers in *K. lactis*, but not to spacers in *S. cerevisiae* (Appendix A). Synthesis of HA-uL4 was induced in *S. cerevisiae* with β-estradiol as above, but before lysis an equal number of *K. lactis* cells harvested from YPD cultures were added to the induced *S. cerevisiae* cells. Thus, if the tagged HA-uL4 protein was transferred during lysis from *S. cerevisiae* precursor particles to the *K. lactis* pre-ribosomes, the anti-HA precipitate would contain *K. lactis* precursors particles, including *K. lactis* transcribed spacers. However, no *K. lactis* ITS1 or ITS2 co-purified with *S. cerevisiae* uL4 showing that uL4 is not exchanged between ribosomal precursor particles during lysis (Figure 5C). As controls we also loaded mixtures of anti-HA pull down and total RNA from the two species. The *K. lactis* ITS1 and ITS2 probes only hybridized to bands containing purified *K. lactis* RNA demonstrating the specificity of the probes used (Figure 5C).

## 3. Discussion

Electron micrographs of yeast rRNA genes associated with RNA transcripts show that rRNA for both the small and large subunits is folded during transcription [6]. During rapid growth of yeast cells 70%–80% of the transcripts are cleaved at the A2 site in ITS1 (Figure 1A) into 40S- and 60S-specific segments when RNA I polymerase has transcribed 25% of the 60S-specific segment [5,6] (Figure 6). This raises a central question: Does the assembly of 60S ribosomal proteins begin before or after the rRNA moieties for the two subunits have been separated? The answer to this question is important, because initial co-assembly of 40S and 60S r-proteins raises the possibility that incorporation of proteins from one subunit could affect the assembly pathway and kinetics of the other subunit. Our data suggest that at least a single 60S r-protein, uL4, binds to the nascent rRNA molecule before A2 cleavage separates the 40S and 60S rRNA moieties. The central results justifying this conclusion are that uL4 is associated with precursor particle(s) that also contain a 40S protein (eS7) and both ITS1 and ITS2.

Our experimental strategy was to characterize ribosomal precursor containing specific r-proteins. This was accomplished by inducing the synthesis of specific HA-tagged r-proteins for a short time followed by immune purification on anti-HA coated beads. We observed the incorporation of the HA-tagged protein into ribosomes using a sucrose gradient. As expected, the tagged r-proteins initially entered precursor particles and were not seen in mature subunits competent to combine into 80S ribosomes and polysomes until about 30 min after induction (Figure 2). Considering that it takes 5–10 min after addition of β-estradiol before accumulation of the tagged proteins raises over the basal level (Figure 1C), this is in good agreement with the fact that it takes 10–20 min to process primary rRNA transcript into mature 18S and 25S rRNA molecules (see e.g., [17,18]). Together these results suggest that the tagged r-proteins go through a normal ribosomal assembly process and, furthermore, that the majority of the tagged proteins initially are in precursor particles, which facilitates purifying precursor ribosomes according to their content of a particular protein. By 15–30 min of induction some tagged protein was also found in complexes sedimenting close to the top of the gradient. We suggest that tagged r-proteins synthesized in excess over other ribosomal form artificial complexes, e.g. because unstructured regions in the free proteins could cause a propensity for aggregation. Alternatively, the slowly sedimenting complexes of r-proteins with their chaperones might accumulate, because the chaperone cannot deliver their freight to ribosomal assembly complexes.

To investigate if some 60S r-proteins are incorporated into the emerging ribosomal assembly intermediates prior to the separation of the precursors for 40S and 60S subunits, we first searched for ribosomal precursor particles containing both 40S and 60S proteins. Particles purified on anti-HA coated beads after induction of HA-eS7 synthesis were found to also contain uL4 (Figure 3B), but only trace amounts of uL22, even though uL22 was efficiently co-precipitated with HA-uL4 in the control experiment (Figure 3A). Inspection of structure models of 90S precursor ribosomes and mature 80S suggests that the failure to precipitate 80S containing HA-eS7 on anti-HA beads is due to obstruction of the access to the N-terminal end of HA-eS7 in 80S ribosomes, but not in the 90S precursor ribosomes (Figure 4). Furthermore, the N-terminal end of uL4 is ready accessed 80S ribosomes explaining the pull down of uL22 after induction of HA-uL4 synthesis. We therefore interpret the results from co-precipitation of r-proteins to indicate that eS7 and uL4 both bind to the nascent rRNA before it is cleaved in ITS1, while uL22 binds after the cleavage.

To interrogate the notion of incorporation of uL4 prior to A2 cleavage, we determined the content of the rRNA internal transcribed spacers (ITS1 and ITS2) in particles pulled down by anti-HA purification. These sequences are markers for precursor ribosomes, because they are degraded during ribosome assembly. Furthermore, ITS1 and ITS2 coexist in early ribosomal precursor particles, but are separated into different particles after A2 cleavage (Figure 6). Accordingly, we determined if ITS1 and ITS2 co-precipitate with HA-uL4. Indeed, we found that both ITS1 and ITS2 co-precipitated with HA-uL4 (Figure 5). The co-precipitation of ITS2 is expected, since the ITS2 probe maps upstream of the C2 cleavage site and the exosome-mediated removal of ITS2 RNA from the 3′ end of 5.8S rRNA occurs late in the nuclear maturation process [19]. The co-precipitation of ITS1 with HA-uL4 confirms that uL4 binds to precursor ribosomes prior to A2 cleavage, because the probe for ITS1 maps upstream of A2 in the part of ITS1 that is associated with the rRNA in precursor 40S subunits (Figure 6). The co-precipitation of both ITS1 and ITS2 with HA-uL4 thus bears out the hypothesis formed from the co-precipitation of uL4 with HA-eS7 (Figure 3).

We also determined the ITS1 and ITS2 content in particles pulled down with HA-uS4. In fact, the autoradiogram in Figure 5A shows bands for both internal spacers. However, only the content of ITS1 increased after induction of HA-uS4, indicating that ITS2 in the immunoprecipitated is due to unspecific association. The presence of ITS1 in the HA-uS4 pull down agrees with the fact that it is not removed by cleavage at site D (Figure 1A) until the 40S precursor has been exported to the cytoplasm [3]. The analysis of internal transcribed spacers in the uS4 pull-down therefore suggests that uS4 is incorporated into the 40S precursor particle after A2 cleavage. However, depletion of uS4 inhibits A2 cleavage and formation of 20S precursor rRNA, suggesting that uS4 binding is a prerequisite for cleavage in ITS1 [20]. One potential explanation reconciling these observations is that uS4 is a trigger for ITS1 cleavage as is illustrated by the broken-line arrow between uS4 and the A2 site in Figure 6. If A2 cleavage therefore follows immediately after uS4 binding, particles harboring both uS4 and ITS2 would not accumulate to detectable levels. Ribosomal particles isolated by pull-down with HA-eL13 also contained only ITS2 and not ITS1, suggesting that this protein also binds after A2 cleavage (Figure 6). Finally, the absence of uL22 in the eS7 pull-down suggests that uL22 is also added after A2 cleavage (Figure 6).

Our contention that uL4 is incorporated before separation of the 40S and 60S assembly pathways makes sense because (i) uL4 binds to the 60S precursor early in its assembly and is present in high amounts in the 90S particle pulled down with antibodies against Rrp5, an assembly factor with function in the earliest steps of ribosome formation [21], (ii) uL4 initially binds the Domain I of the large subunit rRNA, even though it contacts five different rRNA domains of the large subunit rRNA in the mature subunit [22,23], and (iii) the uL4 target in Domain I is within the 25% of the 60S rRNA that is transcribed by the time the rRNA is cleaved at A2. 

Our discussion of the results is based on the model of co-transcriptional ITS1 cleavage. This is justified since we did not see ITS2 in the HA-uS4 precipitate and uL22 was absent in the HA-eS7 precipitate. Furthermore, 70–80% of the rRNA transcripts are cleaved co-transcriptionally in ITS1 [5,6]. Any contribution from post-transcriptional rRNA processing would therefore be modest. We also speculate post-transcriptional processing might occur if uS4 for some reason is not added at the time the RNA polymerase has progressed to the point of cleavage in co-transcriptional processing. If the cleavage was missed, because uS4 did not bind, transcription may have to be completed before the next chance of cleavage appears. However, it is also clear that uS4 binding is not the only parameter affecting the co-/post-transcriptional balance, because poor growth conditions, depletion of at least two different 60S proteins (uL1 and uL4), and inhibition of exonuclease Rat1 also favor post transcriptional processing [4,5,6].

The fact that we only found two proteins that bind before A2 cleavage, and three that do not, could be accidental (Figure 6). However, it is worth noting that while it takes about 20 min to fully process rRNA, the molecule is cleaved at A2 when about half of the precursor 35S precursor (about 3000 nucleotides) has been transcribed. Since rRNA is polymerized at a rate of about 50 nucleotides per second, A2 cleavage occurs already about a minute after transcription start, i.e. very early in the total ribosome assembly process.

In summary, we presented evidence that the 40S protein eS7 and 60S protein uL4 bind to the nascent rRNA prior to separation of the 40S and 60S precursor ribosomes. The significance of this is that 40S protein(s) could affect the folding and assembly of the 5′ 60S rRNA, and, vice versa, 60S protein(s) might affect assembly of the 40S. Therefore, we propose that even though r-proteins are essential only for the assembly of their “own subunit”, binding of r-proteins prior to separation of the subunit precursors could affect pathways and kinetics of the assembly of the ribosomal subunit to which they do not belong. This notion is commensurate with the finding that a deletion in the long loop of the Escherichia coli 50S protein uL4 changes the abundance of pre-16S rRNA (17S), but does not block 30S subunit formation. The uL4 mutation also changes the kinetics of 50S assembly, but does not block 50S formation [24]. Thus, *E.* coli uL4 mutant supports the notion of kinetic interactions between the assembly of the subunits. 

Finally, we note that in slower growing cells, cleavage of the long precursor rRNA into subunit specific moieties occurs more frequently after completion of the transcription [4]. Therefore, more of the r-proteins must bind to the rRNA prior to cleavage in ITS1. Our model thus suggests that in slow growing cells, perhaps including mammalian cells, kinetic interaction of the assembly of the two subunits is more extensive than in fast growing yeast.

## 4. Materials and Methods 

Nomenclature. We used the universal nomenclature for r-proteins [25] with the classic nomenclature added after a slash at the first appearance of the name of a given r-protein.

Strain Construction and Culture Growth. To construct plasmids harboring genes for r-proteins tagged at the N-terminal with hemagglutinin (HA) that can be induced by β-estradiol, we replaced the *lacZ* gene on pLGSD5 [13] with the following DNA segments listed in the direction of transcription: a BamHI restriction enzyme site, an ATG start codon, the HA-tag coding sequence, a SalI restriction enzyme site, the *RPL4A* 5′ UTR sequence lacking the start codon, the *RPL4A* coding sequence, a RsrII restriction enzyme site, the 3′ UTR of the *RPL4A* gene sequence including the stop codon. We then utilized the SalI and RsrII restriction sites to sub-clone other r-protein genes (without the ATG and termination codons) into the plasmid. These plasmids were transformed into DBY12021, which contains a gene for a hybrid transcription factor (GEV) a fusion protein consisting of the galactose DNA binding domain, an estrogen response domain, and the VP16 transactivation domain [15]. Cultures were grown in YPD medium to OD_600_ = 0.8–1 (approximately 2 × 10^7^ cells per mL) and induced by adding β-estradiol to a final concentration of 100 nM. Aliquots of the cultures were placed on ice at the indicated times, harvested by centrifugation and frozen at −80 °C until used for preparation of whole cell lysates.

Sucrose Gradient Sedimentation. Cell pellets from 250 ml cultures were resuspended in 1.25 mL lysis buffer (50 mM Tris-HCl pH 7.5, 30 mM MgCl_2_, 100 mM NaCl, 50 µg/mL cycloheximide, and 200 µg/mL heparin) and lysed by vortexing with glass beads six times for 30 s interrupted with about 3 min incubation periods on ice. An additional 1.25 mL of lysis buffer or gradient buffer (50 mM Tris acetate (pH 7), 50 mM NH_4_Cl, 12 mM MgCl_2_, and 1 mM dithiothreitol was added after lysis. Finally, the lysate was clarified by centrifuging at 10,000 RPM for 20 min, and 10 A^260^ units of lysate were layered on top of a 10–50% sucrose gradient in gradient buffer and centrifuged at 40,000 RPM for 4 h in an SW40Ti Beckman rotor. The gradients where then pumped through a colorimeter with a flow cell and a 254 nm light source while fractions were collected in an ISCO Foxy Jr gradient collector.

Immunoprecipitations. Cell pellets from 250 mL cultures were resuspended in 1.25 mL Pull-Down Lysis buffer (25 mM Tris-HCl (pH 7.5), 0.15 M NaCl, 0.05% Tween 20, 30 mM MgCl_2_, complete protease and phosphatase inhibitor cocktail (Roche), 0.25% Triton 100, and anti-foam B (Sigma)) and vortexed for 15 min at top speed at 4 °C. The samples were then clarified by centrifuging at 10,000 RPM for 15 min, and stored at −80 °C in 300 µL aliquots until used. The samples were then thawed on ice and 20 µL of anti-HA agarose resin (Thermo Scientific PI88836 or 26181) was added then rotated end over end at 4 °C for at least 2 h. The resin was pelleted by centrifuging at 3000 RPM for 5 min at 4 °C and the supernatant was removed. Alternatively, magnetic anti-HA coated beads were used. A sample was taken (FT) and the resin was then resuspended in wash buffer (25 mM Tris-HCl (pH 7.5), 0.15 M NaCl, 0.05% Tween 20, 30 mM MgCl_2_) and pelleted three times. Samples from the first supernatant were taken (Wash). Finally, protein was eluted from the resin by incubating with 1.5× sample buffer for standard sodium dodecyl sulfate poly-acrylamide gel at 95 °C for 5 min. For analysis of RNA co-precipitating with the HA-r-protein, the salt concentration for the wash buffer was optimized for each HA-r-protein by washing with wash buffer containing 0.25, 0.4, 0.5, 0.75, or 0.9 M NaCl. RNA was eluted with 8 M urea at 42 °C for 5 min and fractionated on 1% agarose gels in 90 mM Tris-borate (pH 8.3) buffer. The salt concentration for which most rRNA was eluted from the resin was subsequently used for preparing samples for slot blotting. 

Western blots were performed as described previously [25]. Anti-uL4 serum was prepared for our lab by Covance (Princeton, NJ, USA), anti-HA was purchased from Sigma (H9658), and anti-uL22 was kindly provided by Sabine Rospert (University of Freiburg, Germany).

Slot blotting. RNA samples eluted from the anti-HA beads were deposited onto an Amersham Hybond-N nylon membrane (GE Healthcare Life Sciences, Pittsburgh, PA, USA) using the Minifold II Slot-Blot system (Schleicher & Schuell). The membrane was incubated with pre-hybridization buffer (0.3 M NaCl, 20 nM NaPO_4_ (pH6.8), 2.5× Denhard’s solution (http://cshprotocols.cshlp.org/content/2008/12/pdb.rec11538.full?text_only=true), 10% polyethylene glycol (PEG 6000), 1% sodium dodecylsulfate) for ≥1 h. Then a ^32^P end-labeled probe (5 × 10^6^ cpm) was added and the sample was incubated at 37 °C for >6 h with rotation. The blots were then washed once with oligo wash buffer (0.05 M NaCl, 0.02 M NaPO_4_ (pH 6.8), 0.5% SDS) for 10 min at room temperature. Finally, the blots were exposed to phosphor-imaging screen for >3 h and scanned on a Molecular Dynamics Storm Scanner 860. The images were optimized for clear viewing and quantified using ImageJ.

Hybridization probes

S. cerevisiae ITS1 probe (O1663) CTCTTGTCTTCTTGCCCAGTAAAA G

S. cerevisiae ITS2 probe (O1660) AGGCCAGCAATTTCAAGTTAACTCC

K. lactis ITS1 probe CCCAGTAATCTACTCATTCATAATC

K. lactis ITS2 probe ATAGACTTGACACGCAGCCCTGCTC

Sequence alignments were made with CLC Viewer 8.0

Ribosome structure models were constructed using Chimera 2 [26] using data from The Protein Data Bank (PDB). PDB IDs: 90S 5JPQ [10]; 80S 4V7R [18]; early pre-60S 5Z3G [27]; late (cytoplasmic) pre-60S 3JCT [28].

## Figures and Tables

**Figure 1 ijms-20-02806-f001:**
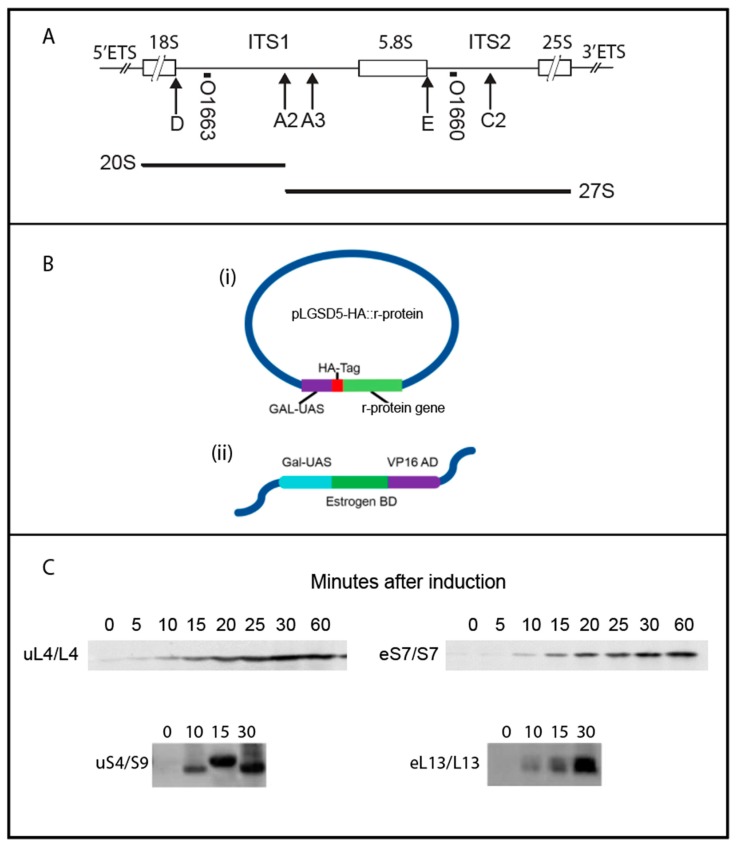
Synthesis of rRNA and hemagglutinin (HA)-tagged ribosomal proteins. (**A**) Map of the rRNA transcription unit with indication of the sequences destined for mature ribosomes (boxes), external (5′ETS and 3′ETS), and internal transcribed spacers (ITS1 and ITS2). Relevant rRNA processing sites are indicated by arrows below the map. Other processing sites are omitted. The short heavy lines immediately below the map show the sequences to which the *S. cerevisiae* ITS1 and ITS2 oligonucleotide probes (O1663 and O1660) hybridize. (**B**) Genetic constitution of strains used. (i) Plasmid (2 µ) carrying a gene for N-terminally HA-tagged ribosomal protein expressed from a *GAL4::CYC1* hybrid promoter. (ii) Gene for a hybrid transcription factor composed of the GAL-DNA binding site, an estrogen response element, and the viral VP16 activating sequence. The gene was integrated in the chromosome. (**C**) Accumulation of HA-tagged uL4, eS7, uS4, and eL13 after addition of β-estradiol was monitored by Western blots probed with anti-HA.

**Figure 2 ijms-20-02806-f002:**
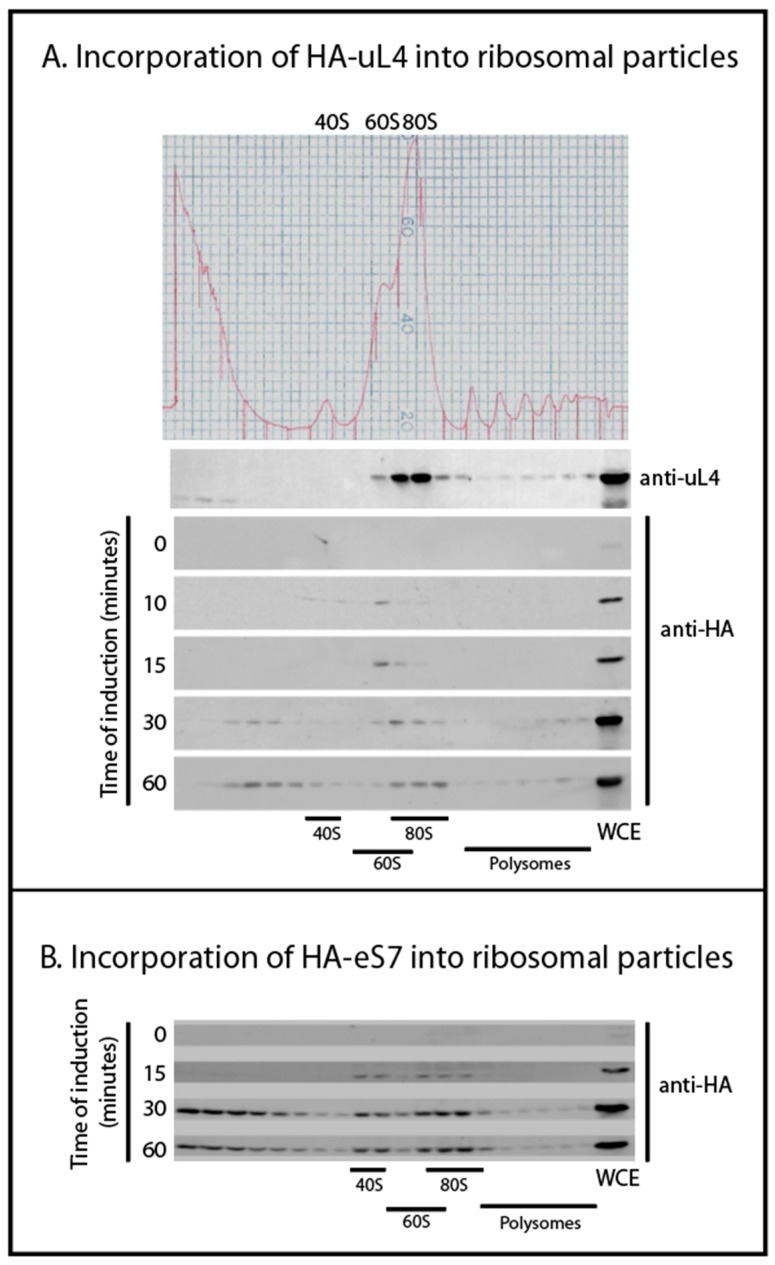
Incorporation of HA-tagged ribosomal proteins into precursor and mature ribosomes. Cells were harvested at different times after inducing the synthesis of HA-uL4 or HA-eS7. Whole cell lysates were fractionated on sucrose gradients and gradient fractions were analyzed on Western blots probed with anti-uL4 or anti-HA. (**A**) Induction of HA-uL4. (**B**) Induction of HA-eS7. Induction times are indicated by each Western blot.

**Figure 3 ijms-20-02806-f003:**
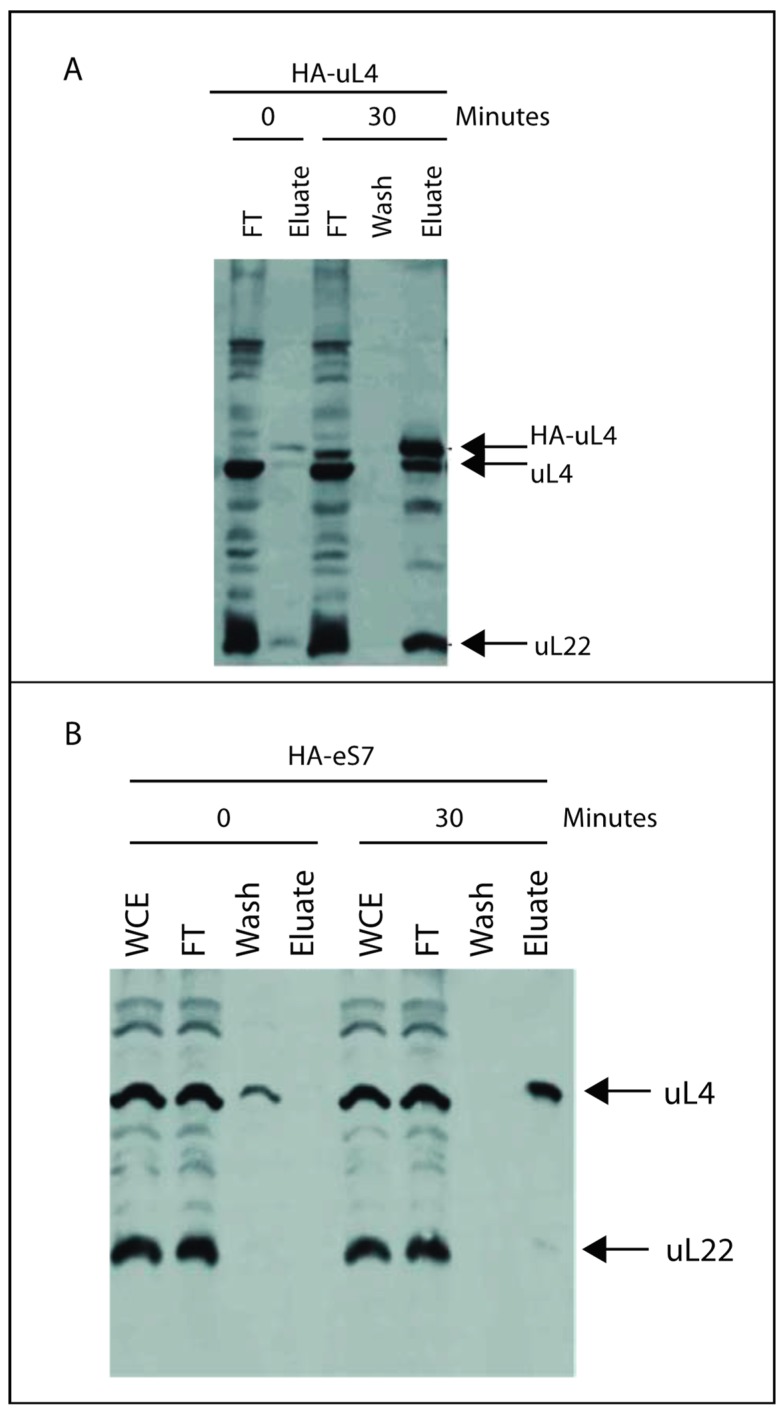
Co-precipitation of r-proteins after induction of HA-uL4 or HA-eS7. Cells were harvested 30 min after induction of HA-uL4 or HA-eS7 synthesis. Ribosomal particles were isolated from whole cell lysates using beads coated with anti-HA. Protein was extracted from the precipitated particles and analyzed by Western blot probed with anti-uL4 and anti-uL22. (**A**) Induction of HA-uL4. (**B**) Induction of HA-eS7. FT: proteins that do not bind to the anti-HA beads (“flow-through”). WCE: Whole cell extract.

**Figure 4 ijms-20-02806-f004:**
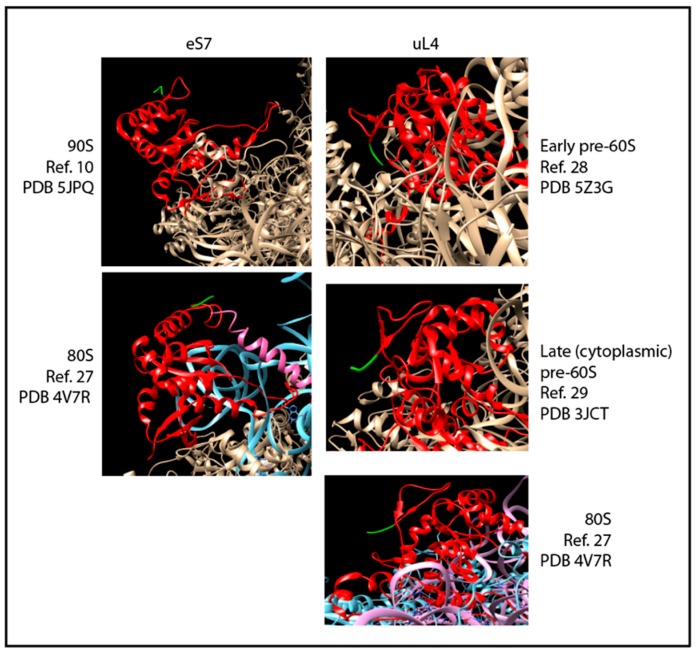
Structure models of the N-terminal ends of eS7 and uL4 in precursor and mature ribosomes. In the left-hand column eS7 is in red with the three N-terminal amino acids in green. Ribosomal protein eL19 in the 80S model is in pink. In the right-hand column uL4 is in red with the three N-terminal amino acids in green. References and PDB IDs for each model are indicated by each panel.

**Figure 5 ijms-20-02806-f005:**
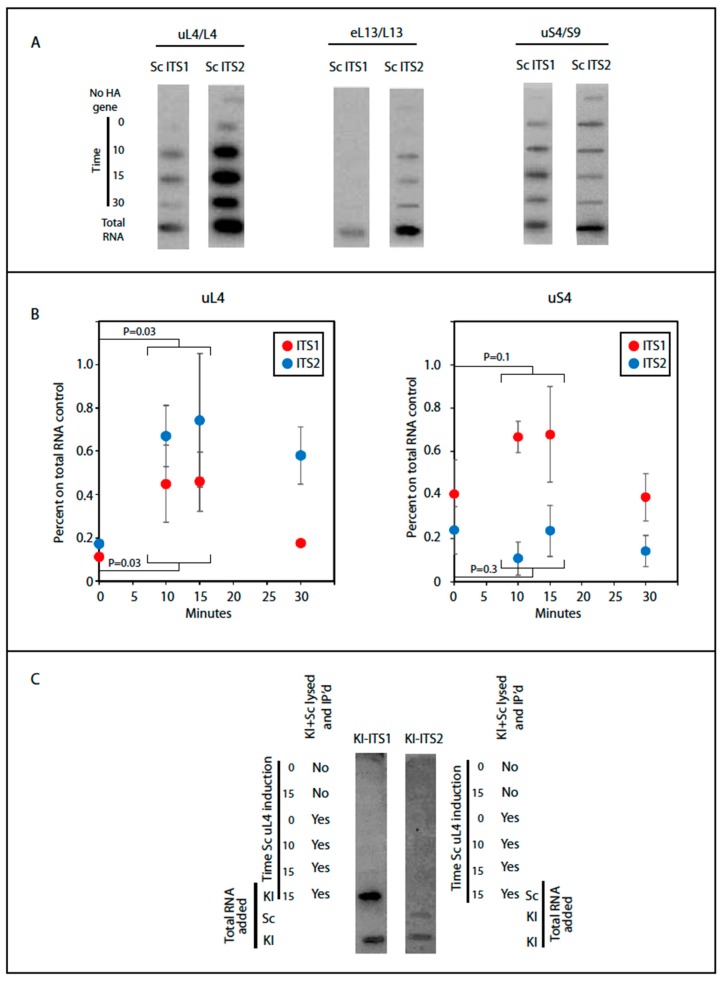
Co-purification of transcribed spacer sequences with HA-tagged ribosomal proteins. Cells were induced for the indicated HA-tagged r-protein for 0, 10, 15, and 30 min. Ribosomal particles were then purified from whole cell lysates using beads coated with anti-HA. (**A**) RNA was extracted from the purified particles and loaded onto a membrane in a slot pattern and probed with ^32^P end-labeled oligonucleotides complementary to segments of ITS1 or ITS2 in *S. cerevisiae* (Sc); see Figure 1A. Slots loaded with total RNA purified from Sc whole cell extracts were used as controls. (**B**) The bands from the uL4 and uS4 immuno-purification in panels A were quantified, normalized to the values for total RNA, and corrected for background from the cells without the plasmid-borne inducible r-protein genes. The results were plotted against time of induction. The combined values for t = 10 and 15 were compared statistically to values at *t* = 0 by *t*-test and the *P* values are indicated. (**C**) Control for unspecific binding of ITS RNA to HA-uL4 containing ribosomal particles after lysis. Sc cells induced for HA-uL4 synthesis for 0, 10, or 15 min were mixed with an equal number of OD_600_ units of *K. lactis* (Kl) and the mix of Sc and Kl cells were lysed together. Lysates were also prepared from Sc cells induced for 0 and 15 min without mixing with Kl cells. HA-containing particles were purified on beads and RNA extracted from the purified particles was used for making a slot blot. Where indicated, the lysate was mixed with total RNA from Kl or Sc. Finally, the slot blot was probed with ^32^P end-labeled oligonucleotides specific for Kl ITS1 or ITS2, see Appendix A.

**Figure 6 ijms-20-02806-f006:**
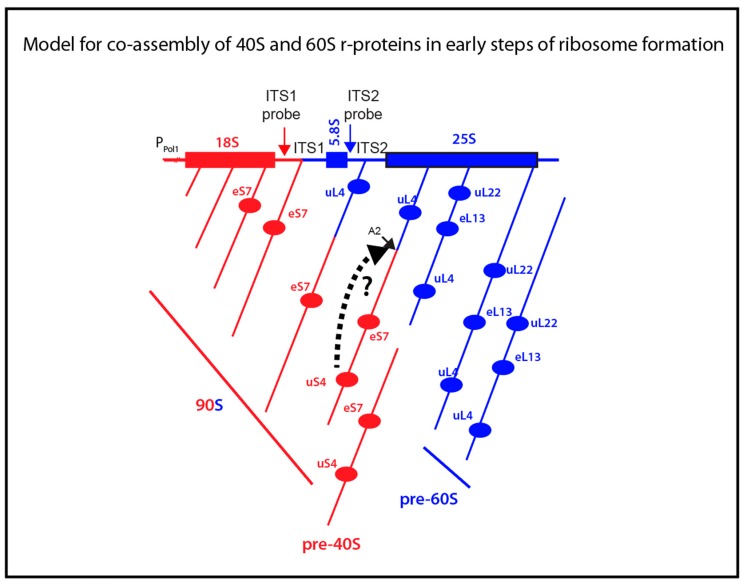
Model for binding of r-proteins to rRNA relative to the time of A2 cleavage separating rRNA segments destined for the small and large subunit. The horizontal map at the top illustrates the rDNA gene. The angled lines below the map illustrates growing rRNA molecules with r-proteins eS7, uS4, uL4, and eL13 binding to the rRNA during transcription. The A2 cleavage is illustrated by separation of the red 40S- and blue 60S-specific rRNA segments. The broken-line arrow from uS4 to the A2 site on the precursor illustrates the hypothesis that uS4 binding stimulates cleavage at A2. RNA and protein belonging to the 40S and 60S subunit are in red and blue, respectively. The type of particle containing the various stages of assembly is indicated below the transcript maps. 40S and 60S specific parts of the rRNA and ribosomal proteins are shown in red and blue, respectively.

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
