# Peer review of "Co-Assembly of 40S and 60S Ribosomal Proteins in Early Steps of Eukaryotic Ribosome Assembly"

_ijms, 2019, doi:10.3390/ijms20112806_

Round 1
Reviewer 1 Report
The authors in the article titled “Co-assembly of 40S and 60S ribosomal proteins in early steps of eukaryotic ribosome assembly” have presented evidences showing association of ribosomal proteins during different time points of ribosome biogenesis. Although this article will be of considerable interest to researchers in the field, it can be further improved by addressing the following issues:
1. Lines 267: the authors have mentioned that “Ribosomal particles purified using HA-uS4 only contained ITS1 and not ITS2”. Similar statement has also been made on lines 272 and 289. Based on figure 4A, I do not think that this statement is true. It could be clearly seen that ITS2 signals are present for “uS4/S9”. In fact, the authors have themselves mentioned on lines 175-176 that “Autoradiographic bands were seen for both the ITS1 and the ITS2 probes with uS4 and uL4…..”
2. Figure 4B: the figure legend should specifically mention what the blue and red levels correspond to.
3. Figure 4C: legend is missing on the top right side and should be inserted like done on the top left side “KI+Sc lysed….”.
4. Line 191: I believe the authors meant “figure 4B” instead of “figure 3B”
5. The authors should rectify the formatting or typograhical issues present in the manuscript like in lines 35, 334, 337, 338 etc.
Author Response
The authors in the article titled “Coassemblynof 40S and 60S ribosomal
proteins in early steps of eukaryotic ribosome assembly” have presented
evidences showing association of ribosomal proteins during different time
points of ribosome biogenesis. Although this article will be of considerable
interest to researchers in the field, it can be further improved by addressing
the following issues:
1. Lines 267: the authors have mentioned that “Ribosomal particles purified
using HAuS4 only contained ITS1 and not ITS2”. Similar statement has also
been made on lines 272 and 289. Based on figure 4A, I do not think that this
statement is true. It could be clearly seen that ITS2 signals are present for
“uS4/S9”. In fact, the authors have themselves mentioned on lines 175176
that “Autoradiographic bands were seen for both the ITS1 and the ITS2
probes with uS4 and uL4…..”
We have expanded the rationale behind the conclusion to which referee 1 refers: After induction of the HA-tagged r-protein the specific concentration (the frequency of ribosomal precursor particles containing the protein) of the tagged protein in ribosomal precursor particles increase. Therefore, the amount of components (such as ITS1 and ITS2) that truly co-precipitate with the tagged protein should also increase. We see this with co-precipitation of uL22 with uL4, uL4 with eS7, ITS1 and ITS2 with uL4 and ITS1 with uS4 (Figures 3 and 4). We do, however, not see an increase of ITS2 in the HA-uS4 precipitate after induction of HA-uS4 (Figure 4B). This difference made us conclude that ITS2 is not specifically co-precipitated with uS4. Therefore, the ITS2 bands in Figure 4 A are not an indication of coprecipitation of ITS1 with uS4. Rather ITS2 for some reason is non-specifically associated with the immune-precipitate. Thus the logical conclusion is that uS4 is added after separation of the 40S and 60S parts of the early pre-ribosome
s
2. Figure 4B: the figure legend should specifically mention what the blue
and red levels correspond to.
We have inserted a color key in the upper right-hand corner of each panel of Figure 4B
3. Figure 4C: legend is missing on the top right side and should be inserted
like done on the top left side “KI+Sc lysed….”.
Corrected. Thank you for pointing this out.
4. Line 191: I believe the authors meant “figure 4B” instead of “figure 3B”
Figure 3B is actually correct, since we are referring to the co-precipitation of uL4 with HA-eS7
5. The authors should rectify the formatting or typograhical issues present
in the manuscript like in lines 35, 334, 337, 338 etc.
Done
Reviewer 2 Report
The authors describe assembly steps or r-protein on nascent rRNA. They show that uL4 binds to rRNA before separation of SSU and LSU pathways and suggest and that uS4 and uL4 may regulate the cleavage at site A2. This is an interesting result. The authors used inducible HA-tagged proteins which are incorporated in mature ribosome and thus in nascent ribosomes.
There are some minor points which have to be considered.
In paragraph 2.3, the authors show co-precipitation of r-proteins co-purified with HA-tagged r-proteins. It is suggested that HA-tag in HA-eS7 protein is not accessible on the surface of mature 40S in 80S ribosome. This explanation is not convincing. In Saccharomyces ribosome structure (PDB 4V88) the N-terminus of eS7 is exposed to the solvent and accessible. A more likely explanation would be that 80S is dissociated during HA pull down experiment. It will be clearer to show that 40S and 60S are dissociated by probing the blot in Fig 3A with an antibody against a 40S protein. It will not change the conclusions of Fig 3B and will rule out the posibbility that 80S is purified in HA pull down experiment
On line 139, it should be written HA-eS7 (instead of HA-uS7)
In Fig 4B, the quantified signals have large errors bars. Can the author provide a statistic test to ensure that observed variations are significant?
The colors for each probe are not indicated. ITS2 seems to be in red and ITS1 in blue. For clarity, it should be best to put uS4 as left panel and uL4 as right panel in order to show the quantification below the corresponding blots. Colors could also be inverted in order to use the same convention as in Fig 5.
In discussion, the sentence "This suggests that initial binding of some r-protein,s for either subunit of rRNA may precede A2 cleavage" (line 221) is not a logic consequence of what is written just before, or it is obscure.
In paragraph beginning at line 267, it would be clearer to introduce the site D cleavage in ITS1 (and show it in Fig 1).
The authors noticed that there is some results in contradiction with the published literature. Sentence in line 277 has some error. "If A2 cleavage therefore follows immediately after A2 cleavage...". It should be 'after uS4 binding".
Author Response
The authors describe assembly steps or rprotein on nascent rRNA. They
show that uL4 binds to rRNA before separation of SSU and LSU pathways
and suggest and that uS4 and uL4 may regulate the cleavage at site A2.
This is an interesting result. The authors used inducible HAtagged Proteins which are incorporated in mature ribosome and thus in nascent ribosomes.
There are some minor points which have to be considered.
In paragraph 2.3, the authors show coprecipitation of rproteins copurified
with HAtagged rproteins.
It is suggested that HAtag in HAeS7 protein is not accessible on the surface of mature 40S in 80S ribosome. This
explanation is not convincing. In Saccharomyces ribosome structure (PDB
4V88) the Nterminus of eS7 is exposed to the solvent and accessible. A
more likely explanation would be that 80S is dissociated during HA pull down
experiment. It will be clearer to show that 40S and 60S are dissociated by
probing the blot in Fig 3A with an antibody against a 40S protein. It will not
change the conclusions of Fig 3B and will rule out the posibbility that 80S is
purified in HA pull down experiment
We should have explained this better, and have now done so in the Results and Discussion in the revised manuscript. However, we respectfully disagree with the Reviewer’s interpretation. The observations cannot be explained by dissociation of 80S ribosomes, because uL4 (a 60S protein) is co-precipitated with HA-eS7. Furthermore, the discrepancy between the co-precipitation of uL22 with HA-uL4, but not with HA-eS7 can only be explained by precipitation of an early ribosome assembly particle containing uL4, but not uL22. This agrees with the finding that uL4/L4 is incorporated into 60S pre-ribosomes before uL22/L17 (Gamalinda et al Genes and Development 2014 doi:10.1101/gad.228825.113.
On line 139, it should be written HAeS7 (instead of HAuS7)
Thanks for pointing this out. The error was corrected
In Fig 4B, the quantified signals have large errors bars. Can the author
provide a statistic test to ensure that observed variations are significant?
t-test results have been added to Figure 4B
The colors for each probe are not indicated. ITS2 seems to be in red and
ITS1 in blue. For clarity, it should be best to put uS4 as left panel and uL4 as
right panel in order to show the quantification below the corresponding blots.
Colors could also be inverted in order to use the same convention as in Fig
5.
Good suggestions. We switched the order of the panels in Figure 4A, introduced color keys in the upper right corner of the panels in Figure 4B. And the colors in Figure 4B are switched to match the colors in Figure 5
In discussion, the sentence "This suggests that initial binding of some rprotein,
s for either subunit of rRNA may precede A2 cleavage" (line 221) is
not a logic consequence of what is written just before, or it is obscure.
We are not sure why the referee finds this illogical or obscure. It seems logical to us that if all of 40S rRNA and 25% of 60S rRNA is transcribed before A2 cleavage, eS7 and uL4 co-precipitate and HA-uL4 precipitates both ITS1 and ITS2 can only be explained by out model that eS7 and uL4 bind before A2 cleavage. We have modified the text in the hope that our thinking is now easier to follow.
In paragraph beginning at line 267, it would be clearer to introduce the site D
cleavage in ITS1 (and show it in Fig 1).
We followed the Reviewer’s suggestion.
The authors noticed that there is some results in contradiction with the
published literature. Sentence in line 277 has some error. "If A2 cleavage
therefore follows immediately after A2 cleavage...". It should be 'after uS4
binding".
Correct. Thank you for pointing this out. We have corrected the error.
Reviewer 3 Report
In this manuscript, the authors investigate the timing of ribosomal protein assembly during ribosome biogenesis. Using pull-down experiments with HA-tagged ribosomal proteins, they showed by sucrose gradient analysis and nothern blot experiments, that the ribosomal protein eS7 and uL4 bind to nascent ribosomes prior 40S and 60S separation by A2 cleavage in ITS1. On the contrary, ribosomal protein uS4 and eL13 bind during or after cleavage. Although this manuscript is of general interest, the overall experiments are over-interpreted and not conclusive.
The authors analyzed the presence of rRNAs and rproteins in total cell extracts, therefore assessing a mixture of mature 80S, mature 40S and 60S and finally pre-ribosomes, the goal of this study. In order to discriminate each category, which is an essential issue in this manuscript, the pull-down results should be confirmed by analysing the nucleic and protein contents from pull-down performed with nuclear extracts, which contain only precursor ribosomes. Therefore, this reviewer cannot recommend this article for publication in ‘International Journal of Molecular Sciences’.
Major concerns:
HA-tagged eS7-pulldown experiments allow the detection of uL4 but not uL22. The authors speculate that this due to an inaccessibility of the HA-epitope in mature 40S and in 80S (p6 – line 140). Therefore the authors interpret this result as a proof that the presence of uL4 in their eS7 pull-down must be due to its presence in pre-ribosomes before the A2 cleavage in ITS1. However, their assumption that the HA epitope of the HA-eS7 is not accessible in the mature 40S and in the 80S is difficult to imagine when looking at the position of eS7 in the yeast 80S structure, indeed eS7 is located on the solvent face and not on the interface between the two ribosomal subunits. Moreover, the lack of accessibility of a ribosomal protein in the 80S is easy to test by western blotting with purified 80S. Altogether, these experiments are not conclusive and the lack of detection of uL22 is a negative result that might be due to alternative reasons.
Therefore, to be conclusive, these experiments should be repeated with nuclear extracts, which contain only pre-mature ribosomal subunits rather than whole cell extracts in order to avoid contamination by mature 80S particles coming from the cytoplasm that renders the interpretations of these experiments challenging.
Minor concerns:
Fig 1c, why the band detected at 15 min is higher than the others?
Where is the HA tag incorporated, N-t or C-t? The authors should also consider the position of the ribosomal proteins that are investigated and show their localisation in the well-known 3D structure of yeast 80S and also in the 90S pre-ribosome, they should also show where the HA-tags are inserted.
Author Response
In this manuscript, the authors investigate the timing of ribosomal protein
assembly during ribosome biogenesis. Using pulldown experiments with HAtagged
ribosomal proteins, they showed by sucrose gradient analysis and nothern blot experiments, that the ribosomal protein eS7 and uL4 bind to nascent ribosomes prior 40S and 60S separation by A2 cleavage in ITS1. On the contrary, ribosomal protein uS4 and eL13 bind during or after
cleavage. Although this manuscript is of general interest, the overall experiments are overinterpreted and not conclusive. The authors analyzed the presence of rRNAs and rproteins in total cell extracts, therefore assessing a mixture of mature 80S, mature 40S and 60S
and finally preribosomes, the goal of this study. In order to discriminate each
category, which is an essential issue in this manuscript, the pulldown
results should be confirmed by analysing the nucleic and protein contents from pulldown
performed with nuclear extracts, which contain only precursor ribosomes. Therefore, this reviewer cannot recommend this article for publication in ‘International Journal of Molecular Sciences’.
We respectfully disagree with the referee on the need for repeating our experiments with nuclear extracts. In each of our experiments we have used criteria that distinguish precursor ribosomes from mature ribosomes. In Figure 3 we have identified a particle that contains eS7 and uL4, but not uL22. This composition can only be explained as a precursor ribosome containing protein from both subunits. In Figure 5 (Figure 4 in the original submission) we have identified precursor particles based on their content of ITS1 and/or ITS2. No mature ribosomes contain these sequences. In short, we cannot see what will be gained by repeating the experiments with nuclear extracts. Furthermore, we note that most, if not all, structures of precursor particles have been determined using immune-purified material from whole cell extracts.
Major concerns: HAtagged eS7pulldown experiments allow the detection of uL4 but not
uL22. The authors speculate that this due to an inaccessibility of the HAepitope
in mature 40S and in 80S (p6 – line 140). Therefore the authors
interpret this result as a proof that the presence of uL4 in their eS7 pulldown
must be due to its presence in preribosomes before the A2 cleavage in ITS1. However, their assumption that the HA epitope of the HAeS7 is not
accessible in the mature 40S and in the 80S is difficult to imagine when
looking at the position of eS7 in the yeast 80S structure, indeed eS7 is
located on the solvent face and not on the interface between the two
ribosomal subunits. Moreover, the lack of accessibility of a ribosomal protein
in the 80S is easy to test by western blotting with purified 80S. Altogether,
these experiments are not conclusive and the lack of detection of uL22 is a
negative result that might be due to alternative reasons.
Therefore, to be conclusive, these experiments should be repeated with
nuclear extracts, which contain only premature
ribosomal subunits rather than whole cell extracts in order to avoid contamination by mature 80S particles coming from the cytoplasm that renders the interpretations of these
experiments challenging.
We are of course aware that eS7 is on the solvent face of the 80S ribosome. However, further examination of the 80S structure shows that the C-terminal end of eL19. As we have now argued in the revised manuscript it is plausible that this obstructs access to the N-terminal end of eS7, especially because HA-eS7 is 9 amino acids longer than the wildtype protein.
Minor concerns:
Fig 1c, why the band detected at 15 min is higher than the others?
We are not sure, but believe that it the 15 minute sample for some reason did not get all the way down in the slot. However, we do not think that this affect the interpretation of the other results in our manuscript
Where is the HA tag incorporated, Nt or Ct?
All proteins were tagged at the N-terminal end. This was in fact shown in Figure 1 and the legend of this figure of the original submission. We we have now incorporated this information in the main text for further clarity.
The authors should also consider the position of the ribosomal proteins that are investigated and show their localisation in the wellknown 3D structure of yeast 80S and also
in the 90S preribosome, they should also show where the HAtags
are inserted.
A new Figure 4 presenting this information has been added to the revised manuscript.
Reviewer 4 Report
In this study Fox and coworkers apply an elegant and simplified approach to investigate the timing of the association of ribosomal protein into rRNA subunit precursors.
The experiment flow is easy to follow, I have only one concern plus some minor points.
In figure 3B uL22 is not detected in the eluate probably because "the HA-epitope is not accessible on the surface of the maure 40S and in 80S ribosomes"
This possibility may in fact explain the observation, however the association with the different HA-RP with precursor rRNA moieties is not clearly assessed in the following figure. In my opinion this could be easily confirmed performing a Norther Blot analysis with ITS1 and ITS2 specific probes at least for 1 time point after induction (e.g. 10 or 15 minutes). This would also rule out that the expression of HA-tagged RPs may have effects altering rRNA processing this conforming the specificity of the slot blot analysis.
Minor points:
Figure legends should include the explanation of the abbreviations (e.g. Figure 3 WCE, FT;) and a more clear title (e.g. current figure 3 title is not easy to understand)
There are several typo throughout the text (e.g. line 90 after 10 minutes; line 94 existence of short induction; line 107 of other ribosomal proteins?)
Author Response
In this study Fox and coworkers apply an elegant and simplified approach to
investigate the timing of the association of ribosomal protein into rRNA
subunit precursors. The experiment flow is easy to follow, I have only one concern plus some
minor points.
In figure 3B uL22 is not detected in the eluate probably because "the HAepitope
is not accessible on the surface of the maure 40S and in 80S ribosomes"
This possibility may in fact explain the observation, however the association
with the different HARP with precursor rRNA moieties is not clearly
assessed in the following figure. In my opinion this could be easily confirmed
performing a Norther Blot analysis with ITS1 and ITS2 specific probes at
least for 1 time point after induction (e.g. 10 or 15 minutes). This would also
rule out that the expression of HAtagged RPs may have effects altering
rRNA processing this conforming the specificity of the slot blot analysis.
We are not quite sure what the referee is after. However, we have now included models of the N-terminal ends of eS7 and uL4 in 90S, 80S and two different pre-60S particles to show the accessibility of the tagged proteins used for immune-precipitation (the N-terminal end of uS4 is not incorporated in the 90S model from Beckmann’s and Hurt’s labs).
We think that our quantification of ITS1 and ITS2 in the uL4 and uS4 immunoprecipitates confirms the association of the internal spacers with the purified precursor particles. Furthermore, Figure 2 shows that the incorporation of HA-eS7 and HA-uL4 into precursor and mature ribosomes is as expected from published assembly kinetics. In our opinion this argue strong against any major effects of the HA-tag on the ribosome assembly
Minor points:
Figure legends should include the explanation of the abbreviations (e.g.
Figure 3 WCE, FT;) and a more clear title (e.g. current figure 3 title is not
easy to understand)
Thank you, this information has been added in the revised manuscript.
There are several typo throughout the text (e.g. line 90 after 10 minutes; line
94 existence of short induction; line 107 of other ribosomal proteins?)
Also thank you for pointing this out. The errors have been corrected.
Round 2
Reviewer 3 Report
In the corrected version of the manuscript that was resubmitted, none of my concerns were addressed by the additional experiments that I requested. In fact, the authors didn’t add any supplementary experiments to their manuscript at all. Concerning the fact that they used whole cell extracts to analyse RNP complexes that are located exclusively in the nucleus, the authors replied that the 90S particle structures have been determined from whole cell extracts. This is indeed correct, however, to solve these structures, the 90S particles have been extensively purified by several steps and carefully characterised before structure analysis and determination to avoid any putative 40S, 60S and 80S contamination. In their manuscript, the authors did not analyse the 90S particles. A careful characterization of the protein complexes obtained from their IP experiments is mandatory, a western blot is far from being conclusive. Moreover, the authors even didn’t repeat the western blot of Figure 1C in which there is a real problem with the size of band uS4/S9 at 15 minutes. Their explanation that the sample did not get all the way down the slot is far from being satisfactory. If this would be the case, I strongly recommend to the authors to repeat this very fast and simple experiment rather than speculating on very unlikely hypothetical explanations. Since none of my concerns have been addressed by additional experiments, this reviewer cannot recommend this manuscript for publication.
Reviewer 4 Report
The Authors convincingly responded to my major concern, I have no additional issues to rise.